# Bike-Sharing Demand Prediction at Community Level under COVID-19 Using Deep Learning

**DOI:** 10.3390/s22031060

**Published:** 2022-01-29

**Authors:** Aliasghar Mehdizadeh Dastjerdi, Catherine Morency

**Affiliations:** Department of Civil, Geological and Mining Engineering, Polytechnique Montréal, Montreal, QC H3T 1J4, Canada; cmorency@polymtl.ca

**Keywords:** bike-sharing, community detection, short-term prediction, LSTM, COVID-19

## Abstract

An important question in planning and designing bike-sharing services is to support the user’s travel demand by allocating bikes at the stations in an efficient and reliable manner which may require accurate short-time demand prediction. This study focuses on the short-term forecasting, 15 min ahead, of the shared bikes demand in Montreal using a deep learning approach. Having a set of bike trips, the study first identifies 6 communities in the bike-sharing network using the Louvain algorithm. Then, four groups of LSTM-based architectures are adopted to predict pickup demand in each community. A univariate ARIMA model is also used to compare results as a benchmark. The historical trip data from 2017 to 2021 are used in addition to the extra inputs of demand related engineered features, weather conditions, and temporal variables. The selected timespan allows predicting bike demand during the COVID-19 pandemic. Results show that the deep learning models significantly outperform the ARIMA one. The hybrid CNN-LSTM achieves the highest prediction accuracy. Furthermore, adding the extra variables improves the model performance regardless of its architecture. Thus, using the hybrid structure enriched with additional input features provides a better insight into the bike demand patterns, in support of bike-sharing operational management.

## 1. Introduction

Public bike-sharing systems were suggested at the beginning of the millennium but have been gaining momentum only in the last decade. The main premise of implementing bike-sharing systems is to promote sustainable mobility in urban areas. They offer a convenient and easy-to-use service for residents for short-distance trips. Moreover, they are capable to improve first/last mile connection to other travel modes, reduce traffic congestion and energy consumption and decrease environmental impacts of daily travels [1,2]. Furthermore, communities who organize a bike-sharing program increase physical activity and encourage remarkable health benefits to the users [3].

Bike-sharing system as a sustainable and affordable travel mode is not without its challenges for both users, e.g., perceived lack of safety [4], and operators. In this context, bike repositioning or rebalancing has been recognized as an important operational challenge. Bike demand is basically non-stationary, meaning that it varies over time and space. The fluctuating demand may cause the uneven distribution of bikes across different stations where some stations may be totally saturated while concurrently others may lack bikes. This can be related to the “tidal flows” of bike-sharing trips, with certain areas in the city encountering a deficit of bike availability [5]. For instance, during the morning rush hour, residential areas generate a high number of commuting trips towards the areas of employment. This could possibly lead to the problem of insufficient bikes in those areas in that period.

The service quality has positive and significant effect to increase the bike-sharing popularity, attract more users, and improve the overall economic performance of bike-sharing systems. The bike imbalance issue may cause reduced service reliability, user dissatisfaction and decrease attraction and user engagement in bike-sharing program [6,7] which may fail to meet the expectations of sustainable transport system implementation [8,9]. Therefore, it is of great importance to understand and predict travel demand to support planning and day-to-day operation of bike-sharing systems. Accurate and reliable trip demand predictions across the city over different times of day allow system operators to better plan bike redistribution and fleet rebalancing. Hence, application of advanced predictive models has recently received a lot of research interest, as revealed by a recent literature review by Albuquerque et al. [10] on Machine Learning techniques’ contributions applied to bike-sharing systems to improve urban mobility.

Like most bike-sharing systems, the one in Montreal faces the operational challenge of redistributing bikes across the stations to meet travel demand. While optimization algorithms can support such operation, they must rely on relevant travel forecasting demand able to anticipate where bikes will be required in the short-term. This is typically the missing component since bike-sharing demand combines both regular and irregular use patterns and fluctuates according to various events. Moreover, the recent COVID-19 pandemic drastically impacted all features of daily travel for all transport modes. It has become even more challenging to anticipate plausible travel behaviors at all forecasting horizons with the high uncertainty related to post-COVID activity systems. In this context, proposing tools able to forecast pickups while accounting for changing spatial-temporal patterns is critical for network operators that must ensure good fit between provided services (shared bike availability) and evolving demand.

The focus of this study is on the short-term prediction, in a 15 min horizon of bike-sharing usage in Montreal. The timespan is selected in such a way that it accommodates for predicting the bike demand during the COVID-19 pandemic using past data. More specifically, this study aims to compare performance of different models fitted to time-series data when the period under analysis includes important disruptions as the one faced during the COVID-19 pandemics. Firstly, this study employs Louvain method to identify and cluster communities in the bike-sharing network to account for the interactions between stations and be less volatile than station-based modeling. Secondly, the study introduces data structure preparation where historical demand, feature engineering, weather conditions, and temporal variables are incorporated. Thirdly, the study employed deep neural networks for short-term travel demand prediction in a bike-sharing system. In this regard, a hybrid model composed of convolutional neural network (CNN) and long short-term memory (LSTM) is suggested to forecast the time series changes of bike usage. Finally, to examine the effectiveness of the proposed structure, the study compares the performance of some competitors using mean absolute error (MAE) and root mean squared error (RMSE) as evaluation measures.

## 2. Related Works

Empirical studies used different types of predictive models for bike-sharing demand forecasting. Historical data coupled with several exogenous inputs, including weather, temporal and spatial information were typically fed into the models.

Caulfield et al. [11] employed a logistic regression model to explore the usage patterns of the bike-sharing scheme, with some related variables such as travel distance, weather conditions and temporal variables. Ashqar et al. [12] compared univariate models of Random Forest (RF) and Least-Squares Boosting with multivariate model of Partial Least-Squares Regression to predict the number of available bikes. As for the input, available bikes at the station and in its surrounding ones, weather conditions and temporal variables were considered. The univariate models resulted in relatively lower prediction errors. Reynaud et al. [13] developed a panel mixed generalized ordered logit model to predict the availability of bikes at stations at the hourly basis. The model accommodates the influence of temporal, meteorological, bicycle infrastructure, built environment and land-use attributes on bicycle availability. In a study by Sohrabi et al. [14], generalized extreme value (GEV) count models were implemented for the prediction of hourly bike pickups and drop-offs at each station while accounting for time-of-day, weather, built environment, infrastructure, temporal, and spatial dependency factors. Sathishkumar et al. [15] employed different machine learning (ML) models to predict hourly rental bike demand in a city through a one-year dataset of historical data, weather conditions and date information. Linear Regression, Gradient Boosting Machine, Support Vector Machine, Boosted Trees, and Extreme Gradient Boosting Trees were trained and validated. Results showed that Gradient Boosting Machine can provide the best and highest R-squared value.

Short-term prediction is essential for the efficient operation of different traffic networks as it provides informed estimates that are predictive in determining the direction of future trends from a few seconds to possibly a few hours. In recent years, deep learning-based prediction models have made significant progress in short-term traffic prediction, including in bike-sharing systems.

Pan et al. [16] suggested a LSTM model to predict the hourly pickup and drop-off demands in different areas of a city using historical data and weather conditions. For efficient rebalancing of bikes among different bike-sharing dockers, Liu et al. [17] suggested a modeling method based on LSTM to predict available bikes in both one-time step and multi-time steps. Historical data, day of the week and time of the day were fed into the model. Wang and Kim [18] used recurrent neural networks (RNNs) models to predict the short-term available number of bikes (i.e., in 1 min, 5 min, and 10 min time intervals) in docking stations with one-month historical data. As a benchmark, RF algorithm was used. They found that RF is less computationally expensive in training whereas LSTM with complex architectures outperforms its competitors by providing better prediction for the longer intervals. Boonjubut and Hasegawa [19] compared different RNN multivariate models to predict bike-sharing demand. The results showed the model’s effectiveness and accuracy can be improved through combining the LSTM and GRU architectures together. In a study by Li et al. [20], a hybrid LSTM based model was proposed to simultaneously predict short-term bike demand for all bike-sharing stations in the entire city, modeled as a whole. The proposed method effectively improved the prediction performance by capturing the spatiotemporal dependencies. In a similar study, Ai et al. [21] proposed a Conv-LSTM model for short-term spatiotemporal distribution forecasting of dockless bike-sharing systems. The results show that the hybrid LSTM architecture achieves higher prediction accuracy compared with LSTM alone.

Building on previous experimentation, this study demonstrates the relevance of a set of deep learning models to forecast bike-sharing demand using historical data covering the COVID-19 period. Prior studies stressed the superior forecasting skill of CNN-LSTM models over conventional recurrent neural network (RNNs) and traditional ML approaches [22,23,24]. Despites the studies cited above using deep learning models, there is a lack of research implementing the hybrid architecture for bike-sharing systems. The study follows the stream of literature on bike-sharing travel demand prediction but extends previous research by proposing a time series hybrid CNN-LSTM model which is also enriched with several exogenous inputs. As suggested by prior studies, feature engineering of historical travel demand [25] and adding other explanatory variables [17,26] could improve the model performance for short-term prediction. Past literature also supports the effect of these variables on bike-sharing user’s behavior. For example, it was found that the bike-sharing services are widely used in the PM period when it comes to Montreal and New York City. Furthermore, the bike-sharing demand distributions on weekdays and weekends are different [27,28,29]. In general, there is a strong positive correlation between good weather and number of bike-sharing trips. Increase in temperature in cold region such as Montreal has positive influences on bike-sharing ridership [28,30], whereas it is negatively affected by precipitation, wind, and snowfall [29,31,32]. By incorporating such variables, this study hence provides a new set of insights into the potential of hybrid deep learning framework to support a more accurate short-term forecasting of bike-sharing demand.

To tackle the challenge of evolving network configuration faced in the Montreal bike-sharing network as well as to account for interactions between stations, the research uses a community detection approach to aggregate stations at a spatial level. Station level prediction and modeling might be a better solution to support bike fleet management in terms of meeting user’s demand at each individual station. However, compered to aggregated stations, the demand predictions can be less accurate because of higher levels of noise in the data [33]. Previous studies predicted bike-sharing travel demand for regionally clustered stations and provided some reasons for this approach instead of station based [16,33,34,35]. For example, there is a possibility of removal, relocation, and addition of bike-sharing stations in urban areas over time. Therefore, analyzing and modeling at spatial aggregation level captures more efficiently the dynamic nature of local travel patterns and provides a better insight for changes in the dynamics of travel demand and flow over time. The increase popularity of dockless bike-sharing system and its emerging implementation is another reason which may change the current operating methods of bike-sharing system in the future, indicating the importance of the bike travel analysis at area-based levels (i.e., modeling regional dynamics) instead of individual stations to address such change [36,37]. Furthermore, as cited by Yang et al. [33], area-based aggregation of bike sharing stations “supports bike fleet management regardless of the scheme type, with sufficient spatial grain to support rebalancing”.

Since the beginning of the COVID-19 pandemic, the impact of this extreme event on urban mobility, in particular changes in activity-travel and mode choice behavior, has been examined by many studies. However, there is only a few studies with a focus on the effect of the pandemic on bike-sharing demand [38,39,40,41,42] which will be further discussed in the Case Study section. Therefore, considering a big difference in bike-sharing trips before and during the pandemic, this study also contributes to the current literature by developing a sophisticated predictive model in which historical data present numerous disruptions due to the COVID-19 pandemic.

Based on what we mentioned earlier, the main gap identified in the literature review can be summarized as:Potential of hybrid deep learning models to support a more accurate short-term forecasting of bike-sharing demand;New method to aggregate bike-sharing stations to address the challenge of evolving network configuration, as well as interactions between stations;Implementing predictive model with important disruptions in historical data because of COVID-19.

## 3. Modeling Approach

### 3.1. Community Detection for Clustering

This study follows the same network wide time series data structure as Wei and Yan [43] who propose a deep learning model to forecast short-term taxi demand, 10 min ahead, in each zone of a city using historical data and other relevant variables. They used a geographic grid to divide the city into several zones.

Instead of using a grid system, this work applied the method of community detection to separate “groups of densely connected” bike-sharing stations. In this context, Louvain algorithm developed by Blondel et al. [44] was selected. This is a heuristic clustering technique attempting to maximize the modularity of a partition of the network, namely community. Modularity is a connectivity measure quantifying the quality of an assignment of nodes to communities. Louvain method is a two-step optimization algorithm which maximizes modularity in a greedy way. First, modularity is locally optimized by allowing each node to move and join the community of its direct neighbors. Therefore, the first step finds a local optimum of modularity. Second, the identified communities are aggregated into super nodes to construct a new network. These steps are repeated iteratively until no increase of modularity is observed. At the end of the algorithm, maximum modularity is achieved, and a hierarchy of communities is created. Louvain method has strong performance and ability to scale well to large graphs and proved its high efficiency compared to other heuristic algorithms [45].

### 3.2. Data Stucture Design

As presented in Matrix (1), the following parameters were considered to form the data structure, for short-term travel demand prediction at the community level:$D_{t}^{c}$ denotes the number of pickups at the *t*th time interval of community c;$D_{t}^{min}$, $D_{t}^{max},\;D_{t}^{mean},\;D_{t}^{0.25q},\;D_{t}^{0.5q},D_{t}^{0.75q}\;\mathrm{and}\;D_{t}^{std}$ denote minimum, maximum, average, 25%, 50%, and 75% quartiles, and standard deviation of the total bike pickups at the *t*th time interval. In Matrix (1), the seven extra features are presented by $EF_{t}^{k}$;$WC_{t}^{l}$ denotes weather conditions with corresponding *l* index at the *t*th time interval;$TD_{t}^{m}$ denotes temporal variables with corresponding *m* index at the *t*th time interval.
(1)$$Data=\left[\begin{array}{ccccccc} D_{0}^{0} & D_{0}^{1} & \ldots & D_{0}^{c} & EF_{0}^{k} & WC_{0}^{l} & TD_{0}^{m}\\ D_{1}^{0} & D_{1}^{1} & \ldots & D_{1}^{c} & EF_{1}^{k} & WC_{1}^{l} & TD_{1}^{m}\\ D_{2}^{0} & D_{2}^{1} & \ldots & D_{2}^{c} & EF_{2}^{k} & WC_{2}^{l} & TD_{2}^{m}\\ \ldots & \ldots & \ldots & \ldots & \ldots & \ldots & \ldots \\ D_{T-1}^{0} & D_{T-1}^{1} & \ldots & D_{T-1}^{c} & EF_{T-1}^{k} & WC_{T-1}^{l} & TD_{T-1}^{m}\\ D_{T}^{0} & D_{T}^{1} & \ldots & D_{T}^{c} & EF_{T}^{k} & WC_{T}^{l} & TD_{T}^{m} \end{array}\right]$$

Matrix (2), $X_{T}^{c}$, is the input for community *c* containing all variables of the *t* historical time intervals considering the *n* time steps, also called a “sliding window”. There is more than one community, hence the matrix serving as input to deep neural network has an extra dimension to embrace all communities.
(2)$$X_{T}^{c}=\left[\begin{array}{ccccccc} D_{0}^{c} & D_{1}^{c} & \ldots & D_{n}^{c} & EF_{n}^{k} & WC_{n}^{l} & TD_{n}^{m}\\ D_{1}^{c} & D_{2}^{c} & \ldots & D_{n+1}^{c} & EF_{n+1}^{k} & WC_{n+1}^{l} & TD_{n+1}^{m}\\ D_{2}^{c} & D_{3}^{c} & \ldots & D_{n+2}^{c} & EF_{n+2}^{k} & WC_{n+2}^{l} & TD_{n+2}^{m}\\ \ldots & \ldots & \ldots & \ldots & \ldots & \ldots & \ldots \\ D_{T-n-1}^{c} & D_{T-n}^{c} & \ldots & D_{T-2}^{c} & EF_{T-2}^{k} & WC_{T-2}^{l} & TD_{T-2}^{m}\\ D_{T-n}^{c} & D_{T-n+1}^{c} & \ldots & D_{T-1}^{c} & EF_{T-1}^{k} & WC_{T-1}^{l} & TD_{T-1}^{m} \end{array}\right]$$

The target is to separately predict the number of bike pickups at each community in the next time interval. This can be presented as;
(3)$$y_{T}=\left[\begin{array}{ccccc} D_{T}^{0} & D_{T}^{1} & \ldots & D_{T}^{c-1} & D_{T}^{c} \end{array}\right]$$

### 3.3. Deep Learning Models

This study mainly focuses on implementing a hybrid CNN-LSTM-based learning framework [46] to predict short-term bike-sharing demand. CNN developed by Lecun et al. [47] is a kind of feedforward neural network which proved its great performance in computer vision, including image classification, object detection, semantic segmentation, etc. CNN specializes in processing data with a grid-like topology and captures spatial dependencies between the grids; however, it can be applied to time series forecasting problems as conducted by [48,49].

CNN models are typically constructed by three different types of layers; namely, convolutional, pooling, and fully connected. The convolution layer as the core building block of the model encompasses a plurality of convolution kernels. The calculation formula is presented in Equation (4). The convolutional and pooling layers are used to reduce the computational complexity. Meanwhile, the fully connected layer is linked to the output as the flattened layer.
(4)$$l_{t}=f_{act}\left(\;x_{t}*k_{t}+b_{t}\right)$$
where $l_{t}$ represents the output value after convolution, $f_{act}\;$ is the activation function,$\;x_{t}$ is the input vector, $k_{t}\;$ is the weight of the convolution kernel, and $b_{t}$ is the bias of the convolution kernel.

LSTM is a special type of RNNs where information processing is sequentially performed by introducing a cell state. Like a memory, LSTM is capable of remembering and forgetting information and then transferring it forward. Therefore, LSTMs are effective at capturing temporal dependencies. The information controlled in the cell state is also carefully regulated by a structure known as the gate. LSTM is associated with three gates including input, forget, and output. Each gate is a neural network layer with its own activation function. Equations (5)–(8) formulate input gate, output gate, forget gate and cell state at the *t*th time interval, respectively.
(5)$$i_{t}=\sigma \left(w_{i}\left[h_{t-1},\;x_{t}\right]+b_{i}\right)$$
(6)$$f_{t}=\sigma \left(w_{f}\left[h_{t-1},\;x_{t}\right]+b_{f}\right)$$
(7)$$o_{t}=\sigma \left(w_{o}\left[h_{t-1},\;x_{t}\right]+b_{o}\right)$$
(8)$$c_{t}=f_{t}⨂c_{t-1}+i_{t}⨂tanh\left(w_{c}\left[h_{t-1},\;x_{t}\right]+b_{c}\right)$$
where $i_{t}$ represents the input gate $\;f_{t}\;$ represents the forget gate, $o_{t}\;$ represents the output gate, $c_{t}\;$ represents the cell state at the *t*th time interval, w represents the weight for respective gate neurons, $h_{t-1}\;$ represents the output of the previous LSTM block at time *t*−1, $x_{t}\;$ is the input matrix at the *t*th time interval, and *b* represents biases for respective gates. ⨂ denotes element-wise multiplication with the same dimensions. *σ* and tanh are the hyperbolic tangent activation functions, respectively. Finally, the information to be transferred to the next period can be denoted as Equation (9).
(9)$$h_{t}=o_{t}\;⨂tanh\left(c_{t}\right)$$

CNN-LSTM architecture, proposed by Shi et al. [49], is a hybrid model integrating a CNN model with an LSTM backend. CNN is used to interpret subsequences of input that together are provided as a sequence to an LSTM model to interpret. More specifically, CNN-LSTM architecture uses CNN layers for feature extraction on input data and combines with LSTM to support sequence prediction.

The combination is thus effective for analyzing and capturing both spatial and temporal dependencies. The convolutional operators add more computing power to the model with fewer parameters than conventional LSTM. Several recent studies employed the hybrid CNN-LSTM model for prediction problems such as traffic speed prediction by Cao et al. [50], taxi demand prediction by Shu et al. [51] and urban expansion prediction by Boulila et al. [52].

## 4. Case Study

This study uses data from the Montreal BIXI bike-sharing system. BIXI only operates from April to the end of October, sometimes mid-November, each year due to unfavorable weather during the winter months in Montreal. Figure 1a shows the spatial distribution of BIXI stations in Montreal in 2020. Publicly available data by BIXI include six attributes for each bicycle trip: origin and destination stations, departure and arrival time, trip duration, and membership details.

The bike-sharing data between 15th April and 31st October from 2017 to 2020 is selected for modeling and analysis. As for 2021, the data from 15th April to the end of May is also available and included. The study uses only the start time and origin station of pickups; similar modelling could be performed for return trips. The dataset includes about 19 million BIXI pickups after data filtering. The bike-sharing demand varies significantly across stations and time of day, as shown in Figure 1b,c. The magnitude of the two heatmaps is the sum of BIXI pickups within two different time intervals throughout September 2020.

Figure 2 compares average hourly number of BIXI pickups per month within the study period, combining weekends and weekdays. There is a clear difference between 2020 and the rest, indicating the significant impact of the COVID-19 pandemic on bike-sharing usage. The frequency of use in time of day before COVID-19 displays clearly a multipeak trend from April to October, which is associated with morning peak, non-commuter peak (12:00 a.m. and 2:00 p.m.), and evening peak.

Contrarily, the BIXI pickups in 2020 do not follow such a trend over the months, in addition to a considerable drop in the frequency of use as shown in Figure 2. Other studies also reported a decrease in bike-sharing usage and change in travel pattern at the city level during the COVID-19 pandemic [53,54]. Nikiforiadis et al. [55] found that people perceive the use of bike-sharing safer than public transit considering the possibility of COVID-19 transmission; however, it is not perceived to be as safe as driving or walking due to less interaction with unknown people.

Figure 3 compares the distribution characteristics of bike usage in 2019 and 2020 estimated by the maximum likelihood method. This work uses lognormal as the prior assumption distribution which is supported as the best fitting to public bike-sharing travel time by the findings of Du et al. [56]. The right-skewed feature of travel time is strong evidence that the travel time in the BIXI system is not entirely random.

The estimated parameters in Figure 3 indicate that the BIXI user’s behavior did not significantly change before and during the COVID-19 pandemic in terms of usage duration. In 2020, the mean, median and mode were calculated to be about 16, 12 and 7 min, respectively, while the counterpart values were 14, 10 and 6 min for 2019. Thus, the bike-sharing usage time turns out to be slightly longer during the COVID-19 pandemic. An increase in the bike-sharing trip duration during the pandemic has also been reported in recent studies [39,40,41].

## 5. Community Structure in BIXI Network

As mentioned in the early section, this study aims at developing a short-term prediction model for bike-sharing demand at origin stations. In this regard, the study decided on 15 min as the prediction horizon time. This was shown by Ashqar et al. [12] to be the most effective time interval for predicting bike availability in bike-sharing systems. After aggregation, historical BIXI data in each station translates into the number of BIXI pickup data sequences.

This study faced issues with the modeling of every station on its own. First, the number of bike-sharing stations in Montreal is not stable over time and increased from 546 in 2017 to 671 in 2021. Second, historical BIXI data analysis and modeling at a per-station basis may show a repetitive and similar pattern and disregard the interactions between neighboring stations. Third, some BIXI stations, usually at the edge of the system as shown in Figure 1b, are less used, which contributes to insufficient bike trip data at the station level. Data sparsity could be an issue for the process of modeling. Moreover, those stations are less prone to supply-demand imbalance since they have few pickups and drop-offs. Therefore, analyzing their time sequence is less of interest.

To address this issue, Louvain method was employed to identify communities in the bike-sharing network. To detect BIXI station community structure, the network was defined as an undirected and weighted graph where the weight of the edge between two connected stations is the number of trips between them. The threshold of 1 was also set to remove connection between stations having no exchange trip. This method of defining edge weights separates the BIXI network into several independent clusters where there are a minimum bike-sharing trips between the clusters i.e., aggregated stations. This is more efficient to support bike fleet management as redistribution can be implemented within each community independently.

2020 was considered as the reference year for detecting communities. In the years beforehand, the communities were determined based on station ID codes. For 2021, since there are some new stations, the Euclidean distance to the nearest station i.e., community decides on their clusters.

This study leveraged an implementation of the Louvain method from Python library “networkx4” and its community detection module. This resulted in 6 best communities of different sizes. Figure 4 shows the distribution of BIXI stations within the communities in 2021 in addition to their sizes over time.

Figure 5 shows the daily BIXI pickups for the 6 communities over the study period. Note that, as mentioned earlier, the BIXI data does not cover the whole year but from mid-April to the end of October. In Figure 5, therefore, the date from 31st December to 15th April is filtered out as there are no BIXI trips available. Noticeably, the bike-sharing demand after COVID-19 is lower in all communities compared to before the pandemic (i.e., before 2019). The magnitude and pattern of BIXI usage are more comparable in the communities of 1, 2, 3 and 4. Community 6 has the least daily BIXI pickups among the rest possibly due to the minimum number of stations. It seems that Community 5 stands on the opposite side with the highest frequency of use.

## 6. Data Preparation for Modeling

### 6.1. Selected Variables

In addition to the historical demand, this study uses weather and time related features to further improve prediction power. Temperature, precipitation, wind speed and relative humidity were extracted from the official Canadian weather website. They refer to $WC_{t}^{l}$ in Matrix (1) where l = 1, 2, …, 4. The 6 dummy variables of time of day (i.e., morning, morning rush, lunch time, afternoon, afternoon rush and evening) and a dummy variable of business day were included as $TD_{t}^{m}$ in Matrix (1) where m = 1, 2, …, 7. It is worth mentioning that the columns of Matrix (1) were standardized through Standard-Scaling before implementing models to avoid the influence of the various data units on the result.

Considering the study period and 15 min window, 81,312 time intervals were obtained in total i.e., 4 × 24 (h) × 847 (day). The time step was set as 96, meaning that one-day consecutive time series data is used for predicting the next 15-min bike pickups. After including weather and temporal variables, the input data was reshaped as (81,216, 114, 6) for conventional LSTM architecture. For the hybrid model, the input data was reshaped to have the required structure as (81,216, 19, 6, 6).

This study also seeks to investigate the effect of COVID-19 through developing a robust model to predict bike-sharing demand during the pandemic using past data. Therefore, the training dataset consists of observations of 2017, 2018 and 2019 i.e., 70%. The validation dataset includes observations between 15th April 2020 and 15th July 2020 to tune the model parameters i.e., 10%. The remaining 20% of time intervals is used as test set.

### 6.2. Model Specification

This study establishes different model configurations and investigates their performance in terms of prediction power. To understand the structure of the neural networks, Figure 6 displays a plot of implemented models; as an example, where Matrix (1) includes historical demand, the engineered features, weather conditions, and temporal variables. In this figure, the layer orders as well as the input and output shape of each layer are presented.

The candidate models can be categorized into 5 groups. Group 1 incorporates two conventional LSTM models being only fed with the historical demand data. Therefore, the model structure of Group 1 is:Figure 6a in which $EF_{t}^{k}$, $WC_{t}^{l}$ and $TD_{t}^{m}$ are excluded from Matrix (1);Instead of the LSTM layer shown in Figure 6a, a single bidirectional LSTM (biLSTM) layer in which $EF_{t}^{k}$, $WC_{t}^{l}$ and $TD_{t}^{m}$ are excluded from Matrix (1).

In Group 1, the layers order is in consistent with Figure 6a; however, the input and output shape of the layers are different as the engineered features, weather conditions, and temporal variables are not incorporated.

Group 2 consists of two CNN-LSTM models being only fed with the historical demand data. The hybrid architecture—namely, “TreNet” [24]—is constructed by a single LSTM layer and two CNN layers. To form the CNN part, two 1D convolutional neural networks are stacked without any pooling layer. The second CNN layer is followed by a Rectified Linear Unit (ReLU) activation function. Each of the flattened output of the CNN’s ReLU layer and the LSTM layer is projected to the same dimension using a fully connected layer. Finally, a dropout layer is placed before the output layer. In this regard, the model structure of Group 2 is:Figure 6b in which $EF_{t}^{k}$, $WC_{t}^{l}$ and $TD_{t}^{m}$ are excluded from Matrix (1);Instead of the LSTM layer shown in Figure 6b, a biLSTM in which $EF_{t}^{k}$, $WC_{t}^{l}$ and $TD_{t}^{m}$ are excluded.

To evaluate the effects of $EF_{t}^{k}$, $WC_{t}^{l}$ and $TD_{t}^{m}$ on prediction performance, Group 3 and Group 4 integrate those explanatory variables and accordingly follow the original format of Matrix (1). Thus, Group 3 includes:Figure 6a in which $EF_{t}^{k}$, $WC_{t}^{l}$ and $TD_{t}^{m}$ are included in Matrix (1);Instead of the LSTM layer shown in Figure 6a, a biLSTM layer in which $EF_{t}^{k}$, $WC_{t}^{l}$ and $TD_{t}^{m}$ are included.

Likewise, Group 4 entails:Figure 6b in which $EF_{t}^{k}$, $WC_{t}^{l}$ and $TD_{t}^{m}$ are included in Matrix (1);Instead of the LSTM layer in Figure 6b, a biLSTM layer in which $EF_{t}^{k}$, $WC_{t}^{l}$ and $TD_{t}^{m}$ are included.

The last group is a univariate ARIMA model with only historical demand as the input. As for its parameters, the lag order (p), the degree of differencing (d) and the order of the moving average (q) were set as 10, 1 and 1, respectively. It is worth mentioning that 6 separate ARIMA models were implemented for analyzing and forecasting time series BIXI pickups for each community.

### 6.3. Model Implementation and Performance Metrics

The proposed time series models were implemented in Keras, the Python deep learning API. To calibrate the model and improve the prediction performance, this work implemented manual grid search for a set of hyper-parameters. All possible combinations of hyper-parameters were tested and ultimately the model with minimum prediction error on test data was selected i.e., Mean Squared Error (MSE). It is worth mentioning that as grid search was time-consuming, we limited the search space and only selected a few numbers of hyper-parameters to be calibrated. Feature maps (CNN filter) were selected from {64, 128, 256}, the size of LSTM later was selected from {50, 100}, the size of dense layer was selected from {32, 64}, batch size from {128, 256} and optimizer was selected from {SGD, Adam}. The grid search resulted in 128 filters in CNN model, 50 hidden nodes in LSTM model, 64 hidden nodes in the dense layer, 256 batch size and Adam as the optimizer. The model training was stopped if the validation loss, MSE, did not show any change after 15 epochs. The ReLU was selected as the activation function to address the gradient exploding or vanishing problem. The regularization method of dropout with a rate of 0.4 was applied before the dense layer to reduce over-fitting.

Our experiment platform is the free account of Google Colab with 1 CPU cores (Intel(R) Xeon(R) CPU @ 2.20 GHz), 13 GB RAM and Tesla T4 GPU. 

To measure the model performance, the two metrics of MAE and RMSE were used. They are mathematically represented as follows;
(10)$${\mathrm{MAE}}_{n}=\frac{1}{T}\sum_{t=1}^{T}\left|y_{t}-{\hat{y}}_{t}\right|$$
(11)$${\mathrm{RMSE}}_{n}=\sqrt{\frac{1}{T}\overset{T}{\underset{t=1}{\sum }}{\left(y_{t}-{\hat{y}}_{t}\right)}^{2}}$$
where *n* is the number of BIXI communities, *T* is the time interval, *y* and $\hat{y}$ are, respectively, the actual and estimated values of BIXI pickups at community level. The average values of MAE and RMSE of the 6 communities were measured to examine and compare the performance of the candidate models.

## 7. Prediction Results

Table 1 shows the results of predictive performance comparison between the five groups of models. The time and number of epochs to train the model are also reported in Table 1. As mentioned earlier, the ARIMA provides the baseline performance in time series analysis. It can be found that the deep learning models substantially outperform the ARIMA which suffers from an overfitting issue. It is worth mentioning that this study also tested multi-headed architecture (i.e., parallel models) for this dataset such as MLP: LSTM and MLP: CNN-LSTM where one head of the model integrates weather conditions and temporal variables as inputs and the other one incorporates the rest. The study modeled different configurations. In all cases, they achieved lower prediction accuracy compared to the suggested structure. Thus, they are disregarded to report here.

Comparing Group (1) with (3) and Group (2) with (4) indicate that enriching models with $EF_{t}^{k}$, $WC_{t}^{l}$ and $TD_{t}^{m}$ contributes to lower test errors and accordingly, improved forecasting accuracy. Thus, incorporating the extra variables in the model increases the forecasting performance of the time series. Comparing Group (1) with (2) and Group (3) with (4) confirms that the hybrid neural network architecture achieved better results over the LSTM. Group (4) outperforms other groups suggesting that the hybrid architecture coupled with the extra variables provides better prediction accuracy than other candidates due to minimum MAE and RMSE values for the test set. This specifies the feasibility of TreNet for time series demand prediction in bike-sharing services. This is in line with the findings of [22,23] supporting the superior performance of TreNet structure over conventional LSTM and traditional ML approaches for trend prediction in time series data.

**Table 1 sensors-22-01060-t001:** Predictive performance comparison.

Model	MAE		RMSE		Epoch	Time ^1^
	Train	Test	Train	Test		
$\mathrm{Group}\;(1):\;\mathrm{Without}\;EF_{t}^{k}$ $,\;WC_{t}^{l}$ $\;\mathrm{and}\;TD_{t}^{m}$						
• LSTM	9.04	7.59	15.08	11.95	26	2 s
• biLSTM	8.98	7.24	14.98	11.38	32	3 s
$\mathrm{Group}\;(2):\;\mathrm{Without}\;EF_{t}^{k}$ $,\;WC_{t}^{l}$ $\;\mathrm{and}\;TD_{t}^{m}$						
• CNN-LSTM	6.09	5.93	8.95	9.11	21	1 s
• CNN-biLSTM	6.04	5.93	9.02	9.32	27	2 s
$\mathrm{Group}\;(3):\;\mathrm{With}\;EF_{t}^{k}$ $,\;WC_{t}^{l}$ $\;\mathrm{and}\;TD_{t}^{m}$						
• LSTM	7.54	5.14	14.00	8.09	29	2 s
• biLSTM	6.57	4.48	11.56	6.86	24	3 s
$\mathrm{Group}\;(4):\;\mathrm{With}\;EF_{t}^{k}$ $,\;WC_{t}^{l}$ $\;\mathrm{and}\;TD_{t}^{m}$						
• CNN-LSTM	3.17	3.00	4.92	4.77	57	2 s
• CNN-biLSTM	3.25	3.09	5.03	4.88	57	3 s
Group (5):						
• ARIMA	8.05	50.95	12.41	61.17	-	762 s

^1^ Time taken for an epoch during fitting.

Figure 7 compares separately the performance metrics of MAE and RMSE for the BIXI communities on the test set. For each BIXI community, Group (4) outperforms other models in the two-evaluation metrics. As presented visually, the model with LSTM layer in Group 4 shows a slightly better accuracy than its counterpart with bidirectional LSTM layer. Therefore, this hybrid architecture is proposed as the model producing best model performance in terms of prediction power.

To comprehend the accuracy of the proposed model visually, the predicted and real values of hourly and daily BIXI trips over test data are shown in Figure 8 and Figure 9 for two selected communities. The percentage error plots are also presented i.e., the difference between predicted and true values, as a percentage of the true value. It is worth mentioning that the study aggregated 15 min BIXI pickup time series into hourly and daily intervals for the sake of visualization. Figure 8 displays the changes in number of BIXI pickups from midnight to 23:00 on 19th August 2020. Note that the percentage error plot related to Community 6 is disconnected at 5 AM as the corresponding real value of BIXI pickups is zero. Both figures show that the predicted values follow their corresponding real ones very closely in both selected communities. However, the predicted values fit the real BIXI pickups in Community 6 better.

The evaluation metrics of prediction on test data are reported in Table 2 considering different time steps i.e., sliding window. Obviously, the hybrid structure enriched with the extra variables (i.e., Group 4) shows the best result compared to other groups. By narrowing down and stretching the sliding window, the advantage of the proposed model is revealed as its prediction performance is less sensitive.

According to the results above, some findings at the technical aspect can be summarized as:
BIXI demand has certain periodicity, and after a detailed analysis, the data clearly have hourly and daily time series periodicity. Therefore, it is essential to choose the last 24 h (i.e., 96 time steps) historical data as the input for forecasting the BIXI pickups of the next 15 min. Comparing Table 1 and Table 2 supports this statement for almost all candidate models as the selected time step provides a better prediction performance on test data;There are many features associated with BIXI pickup demand which may influence the accuracy and efficiency of the short-term prediction. Results show that the accuracies of deep learning models integrating the extra inputs i.e., demand related engineered features, weather conditions, and temporal variables are higher than those of univariate models for short-term prediction of the bike-sharing pickups. Therefore, it is of importance to incorporate the extra features;This study proposed a hybrid CNN-LSTM model to forecast bike pickup demands in different clusters of BIXI stations simultaneously, in which CNN is employed to extract related features from input data, and then LSTM is adopted to support sequence predictions. The results indicate a prominent efficiency of the proposed architecture in terms of prediction accuracy but with a longer training time.

## 8. Discussion and Conclusions

For bike-sharing operators that must, daily, adapt the way they redistribute bikes, having models able to provide good predictions even under important disruptions such as the COVID-19 pandemic is critical. This study has demonstrated the ability of deep learning models to provide efficient and relevant short-term forecast of bike pickup demands while accounting for specificities of predetermined communities of stations. Typical ARIMA models applied to multiple years of time-series data from a large-scale bike-sharing system are easily outperformed by conventional LSTM and CNN-LSTM models. Accounting for variables confirmed to influence bike-sharing demand in previous literature (weather, time of day, type of day) further improves the model performance. Hence, this paper demonstrates that using a hybrid architecture that also included explanatory variables provides the best performance as assessed with MAE and RMSE indicators. In addition to shedding light on the opportunity of using such a model to forecast bike-sharing demand for operational purposes, this paper innovates by using the community concept. Instead of forecasting demand at the system level or relying on an arbitrary zoning system, this research generates groups of stations, called communities, sharing similar desire lines as reflected by historical trips. This further enhanced the modelling demonstration.

While the results are more than conclusive, there is still room for improvement. First, in the BIXI system, stations do not all have the same capacity. Accordingly, the level of usage of a station or set of stations may be influenced by the number of docking stations and this capacity may change over time. This diversity of station configurations may affect the ability of the model to correctly forecast usage; hence, capacity should be included by modelling at the docks level, for instance. Second, some stations are dropped due to changes in the network configuration over time or to unobserved trips in the historical data, so they are not included in the forecasted demand. It may or not be an issue and this would need to be assessed in future research. Relying on the community concept allows the models to be less sensitive to changes in the network; hence, this approach could be used for dockless bike-sharing that rely on operating zones instead of stations. Third, the current models have been tested for pickups only; model performances should also be validated with respect to return patterns since redistribution operations must empty and fill stations so both trip ends are required to meet operational needs. Finally, as concluded by Li and Axhausen [57], there is no universally superior model for short-term traffic demand forecasting. Different ML algorithms/deep learning architectures may provide better performance depending on the nature of the spatial and temporal variables. Specific areas and timestamps play an important role in this regard. Therefore, testing new algorithms and architecture with the same or new dataset will provide more insight to improve short-term prediction accuracy. Testing the algorithms with other levels of temporal resolution (hour, time-periods) could also further demonstrate their ability to assist in both operational and strategic planning.

In addition to the improvements mentioned above, other developments are planned for future research—namely, testing the same modelling approach to taxi demand, free-floating carsharing demand and a combination of these three modes. Key challenges will be the creation of integrated sets of time-series data and adaptation of the community concept to multimodal trips.

## Figures and Tables

**Figure 1 sensors-22-01060-f001:**
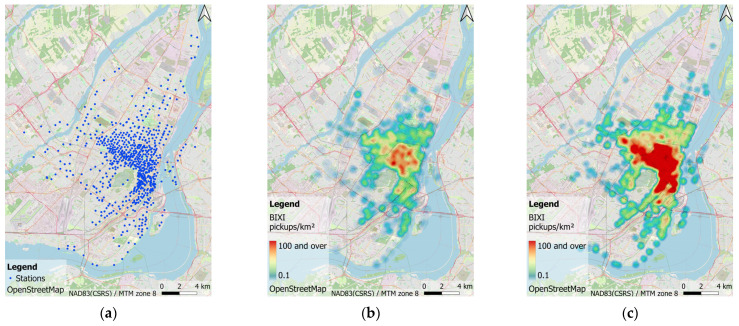
Spatial and temporal distributions of BIXI stations and pickups in Montreal. (**a**) Spatial distribution of BIXI stations; (**b**) Cumulative BIXI pickups heatmap, 6:00–7:00 September 2020; (**c**) Cumulative BIXI pickups heatmap, 14:00–15:00 September 2020.

**Figure 2 sensors-22-01060-f002:**
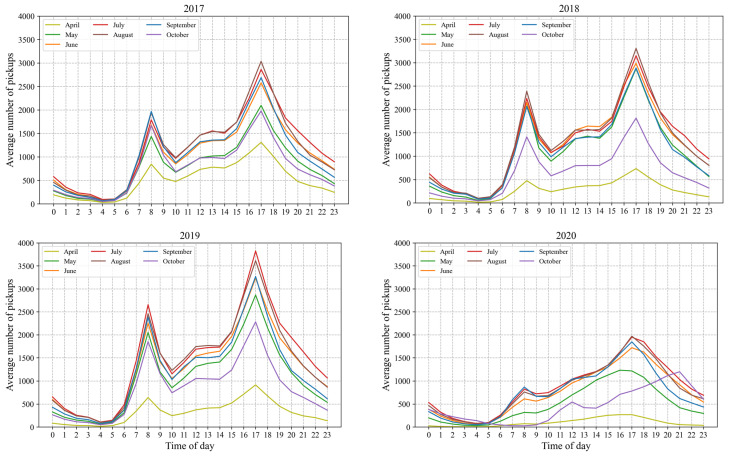
Frequency of use in time of day, per month, from 2017 to 2020.

**Figure 3 sensors-22-01060-f003:**
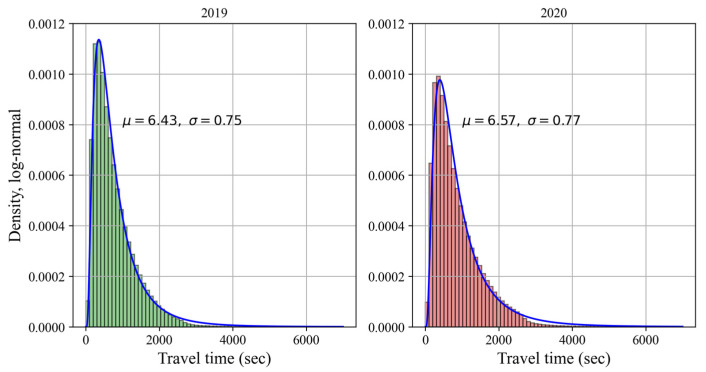
Probability density functions for travel time before and during COVID-19.

**Figure 4 sensors-22-01060-f004:**
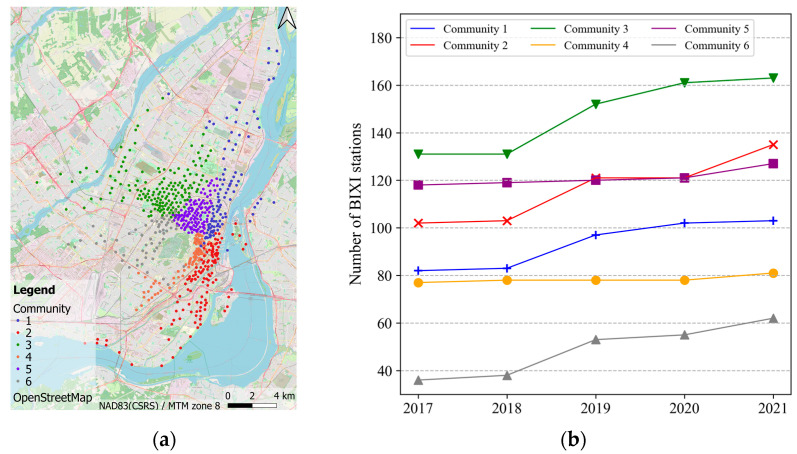
BIXI communities in Montreal. (**a**) Spatial distribution of BIXI communities; (**b**) Changes in the number of BIXI stations in each community over time.

**Figure 5 sensors-22-01060-f005:**
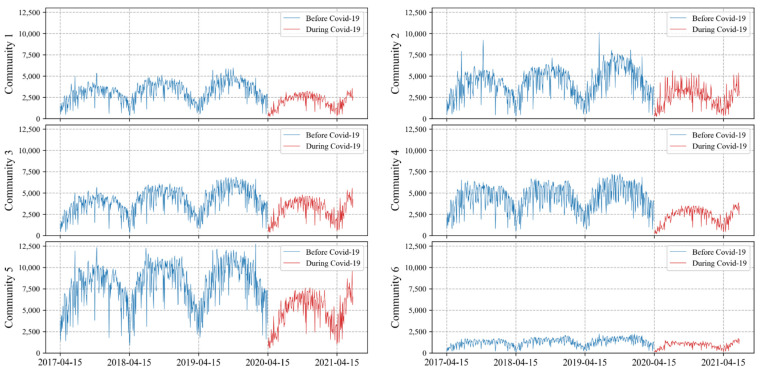
Daily pickups for different BIXI communities over time.

**Figure 6 sensors-22-01060-f006:**
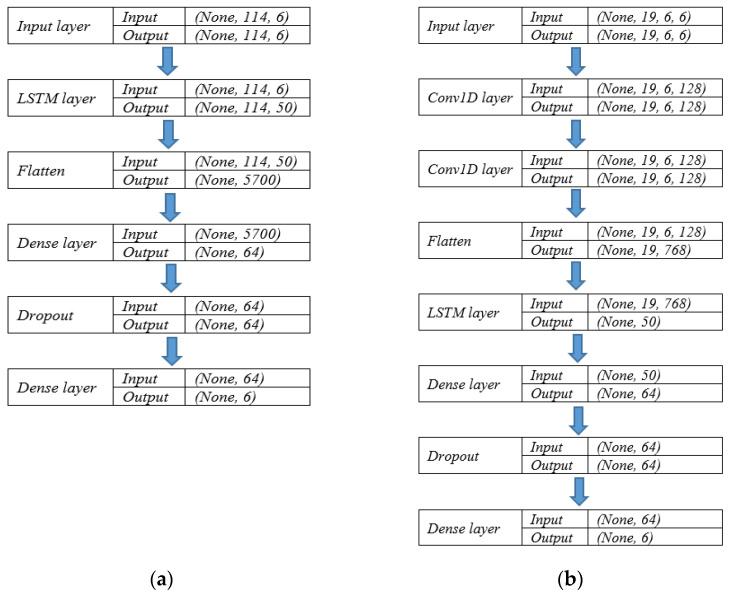
Model configurations. (**a**) conventional LSTM; (**b**) CNN-LSTM.

**Figure 7 sensors-22-01060-f007:**
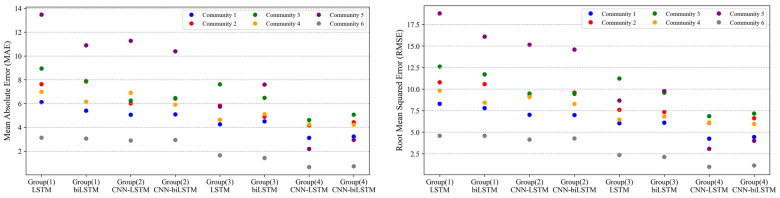
Model performance metrics on test set for individual BIXI community.

**Figure 8 sensors-22-01060-f008:**
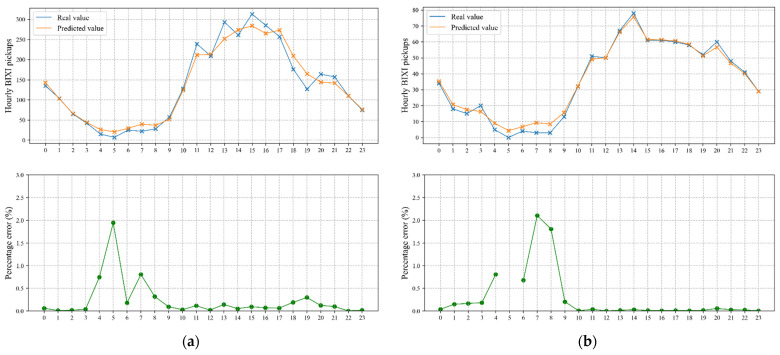
Real and predicted time series as well as the corresponding percentage error of two selected communities on test data for the selected date 19th August 2020. (**a**) Community 3; (**b**) Community 6.

**Figure 9 sensors-22-01060-f009:**
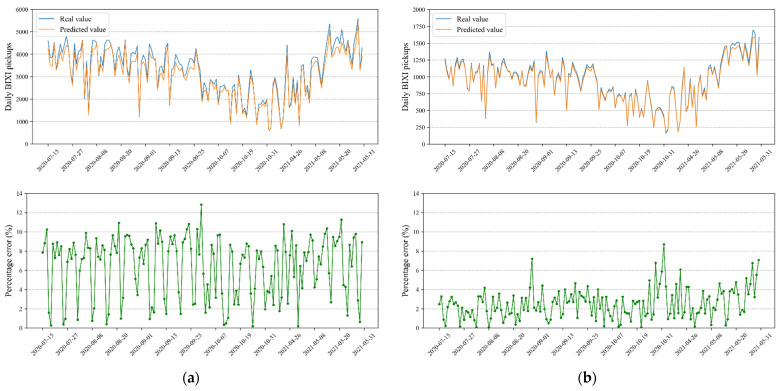
Real and predicted time series as well as the corresponding percentage error of two selected communities on test data. (**a**) Community 3; (**b**) Community 6.

**Table 2 sensors-22-01060-t002:** Evaluation metrics of prediction on test data considering different sliding windows.

Model	24 (6 h)		48 (12 h)		192 (48 h)		672 (1 Week)	
	MAE	RMSE	MAE	RMSE	MAE	RMSE	MAE	RMSE
Group (1):								
• LSTM	7.47	11.18	8.35	12.46	7.04	10.91	15.62	20.42
• biLSTM	7.47	11.08	7.56	11.62	7.14	10.83	20.74	27.41
Group (2):								
• CNN-LSTM	5.74	8.97	5.92	9.31	6.86	9.89	6.27	9.54
• CNN-biLSTM	5.98	9.47	6.23	9.37	6.81	10.11	5.81	9.01
Group (3):								
• LSTM	5.51	9.29	6.46	9.99	5.42	8.97	6.62	9.97
• biLSTM	4.52	7.16	5.43	7.93	10.32	14.42	21.41	28.11
Group (4):								
• CNN-LSTM	3.28	5.02	3.24	5.08	3.12	4.81	3.08	4.74
• CNN-biLSTM	3.34	5.09	3.25	5.03	3.35	5.12	3.53	5.16
Group (5):								
• ARIMA	50.94	61.16	50.94	61.16	50.95	61.17	51.00	61.22

## Data Availability

Historical bike-sharing data in Montreal is accessible through https://bixi.com/en/open-data; accessed date: 15 June 2021. Weather information are available by the official Canadian weather website i.e., www.climate.weather.gc.ca; accessed date: 15 July 2021.

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
