# Peer review of "Bike-Sharing Demand Prediction at Community Level under COVID-19 Using Deep Learning"

_sensors, 2022, doi:10.3390/s22031060_

Round 1

Reviewer 1 Report

The paper presents a method for predicting bikesharing demand at community level using hybrid CNN-LSTM neural network. The title promises the inclusion of pandemic influence on the prediction setup but this skiped in the paper.

Section 2 Related works, mentions bibliography items [37-41] related to effect of pandemic on bikesharing demand but does not discuss their relevance for the essence of the paper.

Section 3 Modeling approach, gives a number of formulas describing some model but the justifiation and idea of the model is not presented.
What is to be modelled and how it is related to the pandemic?
What are the assumptions for the construction and limitations of the model?

Section 6.2 Model specification, presents a number of NN configurations without noting the purpose of chosing such solutions. A univariate ARIMA model is assumed as the reference but no specification is given of its parameters.

Section 7 Prediction results, Table 1 shows results using 4 or 5 significant digits which may not be justified because the number of pickups - I presume in a period of one hour, because it is not clearly stated in the description of the results - does not exceed a few hundreds.
Using samples of this size one commits an error in the range of percents so the results cannot have more than two significant digits. Table 2 may also contain unjustified value notations.

Section 8, Discussion and conclusions, what is the significance of the results of the study for the operation of bicycle shaing entities? How can the results be applied to dockless bike sharing?

Author Response

We really appreciate the reviewer for dedicating the time and effort for helping us improving the second version of the manuscript. We found the comments constructive and highlighted them on the revised attached version. The answers to the comments are provided in the attached file.

Reviewer 2 Report

The following are my conclusions:
- the paper describes an interesting topic;
- the contributions of the paper are based on realistic and referenced assumptions;
- the problem in the manuscript is well defined, and the objectives are clear;
- the paper adequately put the progress it reports in the context of previous works, representative referencing and introductory discussion
- the conclusions and potential impacts of the paper are made clear. 

Increase the bibliography:

bibliographic suggestions

A Novel Acceleration Signal Processing Procedure for Cycling Safety Assessment
E Murgano, R Caponetto, G Pappalardo, SD Cafiso, A Severino
Sensors 21 (12), 4183

Author Response

We really appreciate the reviewer for dedicating the time and effort for helping us improving the final version of the manuscript. Thank you once more for your positive and encouraging comment and pointing out the contribution of the paper. We have added the suggested research in the paper. It is highlighted in the attached revised version.

Reviewer 3 Report

A well-researched and written manuscript. The overall findings are useful for various transportation stakeholders. Few suggestions to improve the paper:

  • The related work section needs to be more critical. For example, similar research has been carried in the US. It is suggested that authors also review some of these research and their findings and thus highlight the concerning GAP.
  • The overall benefit of the suggested deep learning is not convincing. Yes, this tool is useful and as shown in this paper can be advantageous in solving various transportation planning problems. But its use in this paper needs more conviction. Thus more discussion about pattern recognition and calibration is required.
  • Also, consider providing some of the limitations of this research. For example, the algorithms can be modified to show different results. So, how do we ensure that our results are unbiased. This can be rectified by suggesting more parallel examples and investigating their overall comparisons.

The above inclusion would improve the paper significantly. 

Author Response

We really appreciate the reviewer for dedicating the time and effort for helping us improving the second version of the manuscript. Thank you once more for your positive and encouraging comment and pointing out the contribution of the paper.

The answers to the comments are provided below.

  1. The related work section needs to be more critical. For example, similar research has been carried in the US. It is suggested that authors also review some of these research and their findings and thus highlight the concerning GAP.

Thanks for this comment. We have added a new paragraph at the end of the “2. Related Works” to summarize the main contributions and gap in the literature. It is highlighted in the attached revised version.

  1. The overall benefit of the suggested deep learning is not convincing. Yes, this tool is useful and as shown in this paper can be advantageous in solving various transportation planning problems. But its use in this paper needs more conviction. Thus, more discussion about pattern recognition and calibration is required.

Thanks for this constructive comment. We have added two new paragraphs. One at the end of the “7. Prediction Results” section to elaborate more on the suggested model structure and input data.  For the model calibration, we added some explanation at the “6.3. Model Implementation and Performance Metrics” section at the beginning of the paragraph. It is highlighted in the attached revised version.

  1. Also, consider providing some of the limitations of this research. For example, the algorithms can be modified to show different results. So, how do we ensure that our results are unbiased. This can be rectified by suggesting more parallel examples and investigating their overall comparisons.

We agree with the reviewer’s comment. We have added a new limitation part in the “8. Discussion and Conclusion” to address the comment.  It is highlighted in the attached revised version.

Reviewer 4 Report

This work proposes a hybrid model composed of convolutional neural network and long short-term memory for forecasting 15 minutes ahead the shared bikes demand in Montreal. Historical travel demand and other variables (weather conditions, time of day, type of day) are used as an input, and the Louvain algorithm is used for identifying six communities in the bikesharing network. Results show that the proposed deep learning model outperform significantly the ARIMA one and allows predicting bike demand during the COVID-19 pandemic.

The research was carefully designed, the conclusions are supported by the results, and the provided information is relevant for the knowledge field. The manuscript could be considered for publication. The author could consider removing the first paragraph in the Conclusion section since that information is better suited for an introduction.

Author Response

We really appreciate the reviewer for dedicating the time and effort for helping us improving the second version of the manuscript. Thank you once more for your positive and encouraging comments and pointing out the contribution of the paper. We agree with the reviewer’s comment. The mentioned paragraph has been moved to the “1. Introduction” section. It is highlighted in the attached revised version.

Round 2

Reviewer 1 Report

version 2 file sensors-1491748-coverletter

All minor deficiences are corrected.

The authors did not introduce corrections in the paper which remedy the basic deficiency that is the lack of presenting the influence of COVID-19 on bikesharing prediction demand.

The following problems are not elaborated:

1) Incorporation of pandemic parameters into the prediction setup that is:
- parametrisation of factors for preventing the spread of pandemic and being introduced by the community administration,
- parametrisation of factors for the pandemic spread,
- parametrisation of factors changing the maintenance requirements of the bike fleet due to pandemic.  
This will enable the extraction of factors which have potential effect on bike sharing and which are directly related to the pandemic. Apriopriately chosen measures representings these factors wil constitute missing inputs of the designed neural network responsible for the pandemic impact on bike sharing.

2) Correction of the predition period to comply with the dynamics of the pandemic. Covid pandemic "precautions" are not introduced on a 15 min basis but are announced at least a few days beforehand. The argument of Ashqar et al. [12] does not apply to prediction distorted by a pandemic.

3) Definition of the basis for the contruction of the prediction model that is:
- defintion of the domain (what is beig modelled),
- assumptions for the construction of the model,
- limits of applying the constructed model.
Tools for construction of a model can only be chosen when such a basis is clearly defined.

Without elaboration of the listed problems the paper cannot be considered as related to COVID-19. 

Reviewer

Author Response

We really appreciate the reviewer for dedicating the time and effort for helping us improving the second version of the manuscript.

Reviewer 1: Comments and Suggestions for Authors

The authors did not introduce corrections in the paper which remedy the basic deficiency that is the lack of presenting the influence of COVID-19 on bikesharing prediction demand.

The following problems are not elaborated:

1) Incorporation of pandemic parameters into the prediction setup that is:

  • parametrisation of factors for preventing the spread of pandemic and being introduced by the community administration,
  • parametrisation of factors for the pandemic spread,
  • parametrisation of factors changing the maintenance requirements of the bike fleet due to pandemic.

This will enable the extraction of factors which have potential effect on bike sharing and which are directly related to the pandemic. Appropriately chosen measures representing these factors will constitute missing inputs of the designed neural network responsible for the pandemic impact on bike sharing.

It seems clear at this point that the research objectives have not been understood. Our research is not about modelling the impacts of the pandemics on bikesharing usage, it is about comparing performance of models fitted to time-series data when the period under analysis includes important disruptions as the one faced during the COVID-19 pandemics. Since we will not change the objective of our research and the use for bikes sharing operators, we have clarified the objective at the beginning of the paper to make sure that the focus is clear and that it does not induce and understanding that we are trying to do something else.

However, to make the research objective clearer, we have added a new sentence in the Introduction section which is “More specifically, this study aims to comparing performance of different models fitted to time-series data when the period under analysis includes important disruptions as the one faced during the COVID-19 pandemics.”

2) Correction of the perdition period to comply with the dynamics of the pandemic. Covid pandemic "precautions" are not introduced on a 15 min basis but are announced at least a few days beforehand. The argument of Ashqar et al. [12] does not apply to prediction distorted by a pandemic.

We appreciate the reviewer for giving this comment. The title and objective of the paper is about short-term prediction, e.g.,15 minutes, 1 hour or a day. We believe that changing the time window will not add any significant contribution to the paper aim and scope. It worth mentioning that at the early stage of this research we implemented a daily prediction using the same modeling architectures. The suggested architecture also provided the best result.

 3) Definition of the basis for the construction of the prediction model that is:

  • definition of the domain (what is being modelled),
  • assumptions for the construction of the model,
  • limits of applying the constructed model.

Tools for construction of a model can only be chosen when such a basis is clearly defined.

We believe that enough information including data structure and deep learning formulas are provided in Section 3.2 Data Structure Design. The point of the reviewer is not clear to us.